# Genetic Variant of the Canine *FGF5* Gene for the Hair Length Trait in the Akita: Utility for Hair Coat Variations and Welfare in Conservation Breeding

**DOI:** 10.3390/genes16080927

**Published:** 2025-08-01

**Authors:** Shinichiro Maki, Md Shafiqul Islam, Norio Kansaku, Nanami Arakawa, Akira Yabuki, Tofazzal Md Rakib, Abdullah Al Faruq, Osamu Yamato

**Affiliations:** 1Laboratory of Clinical Pathology, Joint Faculty of Veterinary Medicine, Kagoshima University, Kagoshima 890-0065, Japan; k6993382@kadai.jp (S.M.); si.mamun@ymail.com (M.S.I.); k9829543@kadai.jp (N.A.); yabu@vet.kagoshima-u.ac.jp (A.Y.); rakibtofazzal367@gmail.com (T.M.R.); faruqabdullahal103@gmail.com (A.A.F.); 2Faculty of Veterinary Medicine, Chattogram Veterinary and Animal Sciences University, Khulshi, Chattogram 4225, Bangladesh; 3Laboratory of Animal Genetics and Breeding, Department of Animal Science and Biotechnology, School of Veterinary Medicine, Azabu University, Sagamihara 252-5201, Kanagawa, Japan; kansaku@azabu-u.ac.jp

**Keywords:** Akita Inu, canine *FGF5* gene, hair length trait, incompletely dominant, dog, welfare

## Abstract

**Background/Objectives**: Variations in hair length are observed in many dog breeds, as determined by the canine *FGF5* gene. Long-haired Akitas, which are disqualified under breeding standards of Akitas, are sometimes born to short-haired parents and may have been subjected to treatments compromising animal welfare. Here, we aimed to identify an *FGF5* variant associated with hair coat variations in Akitas in Japan, and to assess how welfare of this breed can be improved by carefully planned breeding. **Methods**: DNA samples were obtained from 60 Akitas in 2021 (modern Akitas) and 73 Akitas in the 1970s and the 1980s (classic Akitas). Sanger sequencing was performed on all exons and exon–intron junctions of the *FGF5* gene to determine the causative variant of long hair in Akitas. A real-time PCR assay was developed to genotype *FGF5*:c.578C>T in modern and classic Akitas. Using 54 dogs from modern Akitas, scores (1 to 10) of hair length were compared among the three genotypes (C/C, C/T, and T/T). **Results**: Sanger sequencing revealed that the canine *FGF5*:c.578C>T variant was associated with long hair in Akitas in Japan. Genotyping revealed that the frequency of the mutant T allele was 0.350 in modern Akitas, which was significantly higher (*p* < 0.001) than in classic Akitas (0.212). The three genotypes were not in Hardy–Weinberg equilibrium (HWE) in modern Akitas but were in HWE in classic Akitas. There were significant differences in hair length scores among the three genotypes (*p* < 0.001) and between the C/C and C/T genotypes (*p* < 0.005). There was no significant difference in the scores between male and female dogs. **Conclusions**: This study revealed that a causative variant that determines the long hair trait of Akitas in Japan was the *FGF5*:c.578C>T variant, which was inherited in an incompletely dominant manner. Akita dog breeders were more likely to select heterozygous C/T dogs based on the appearance of the hair coat for breeding dogs with an ideal fluffy hair coat. This might result in a high mutant T allele frequency and the production of undesired long-haired Akitas with T/T, which may create welfare problems. Genetic testing for this variant is necessary to improve welfare and conserve the Akita breed.

## 1. Introduction

Hair coat variation is an important morphological trait in dogs, as it characterizes different domestic dog breeds. Combinations of genetic variations in the three genes—namely, fibroblast growth factor-5 (*FGF5*), R-spondin-2 (*RSPO2*), and keratin-71 (*KRT71*)—determine major hair coat phenotypes, including short, wire, wire and curly, long, long with furnishings, curly, and curly with furnishings [1].

Among hair coat phenotypes, long hair is an attractive trait in dogs, and its variations have been observed in many dog breeds [2,3]. Hair length is determined by the canine *FGF5* gene, which is composed of three exons and is located on the dog chromosome (CFA) 32 [1,2] (Figure 1). According to the Online Mendelian Inheritance in Animals (OMIA) [4], five genetic variants or mutations in the canine *FGF5* gene have been associated with canine long hair (OMIA:000439-9615). These include NM_001048129.1:c.284G>T (NP_001041594.1:p.C95F) [2], c.368-11T>A, c.556_571del (p.A186Tfs*71), c.559_560dup (p. R188Afs*75), and c.578C>T (p.A193V) [5] (Figure 1); these mutations determine long-hair phenotypes in an autosomal recessive mode in multiple dog breeds [4]. Similarly, five genetic variants or mutations in feline *FGF5* have been reported to be associated with long hair in cats (OMIA:000439-9685) [6,7,8,9]. *FGF5* genetic variants associated with long hair have been reported in other species, including rabbits [10], guinea pigs [11], golden hamsters [12], goats [13], donkeys [14], Arabian camels [15], and llamas [16].

Traditional Japanese dog breeds, including Shiba, Kishu, Shikoku, Kai, Hokkaido, and Akita, are approved by Nihonken Hozonkai (NIPPO), a specialized kennel club for traditional Japanese dog breeds [17,18], and the Japan Kennel Club, which is certified by the Federation Cynologique Internationale [17,19]. The Akita, also known as the Akita Inu or Akita Ken, has also been approved by the Akita Inu Preservation Society (Akita Inu Hozonkai, AKIHO), a specialized kennel club for the Akita breed [20]. The Akita is categorized as the only large Japanese dog breed [18,20] and is currently bred in other countries, including the United States, according to local breeding standards [18].

The breed standards for these Japanese breeds do not include any long hair variations; however long-haired Akitas (Figure 2) are sometimes born to short-haired parents. The long-haired Akitas do not meet the standards for both the Japanese dog breeds in NIPPO [18] and the Akitas in the AKIHO [20] and are disqualified at dog shows exclusive for the Akita breed. Therefore, long-haired Akitas may have been subjected to treatments that violate animal welfare. However, they have recently gained popularity among enthusiasts and fanciers.

Previously, researchers from Germany reported that the causative variant of long hair in Akitas was *FGF5*:c.578C>T (p.A193V), using ten long-haired (genotype: T/T), four short-haired (genotype: C/T), and three short-haired (genotype: C/C) Akitas in Germany [5]. This variant was also found in Samoyeds and Siberian Huskies and was inherited in association with the hair length phenotype of these three dog breeds following an autosomal recessive inheritance pattern [5]. However, the relationship between *FGF5* variants and hair length phenotypes in the Akita population in Japan has yet to be investigated.

This study aimed to identify an *FGF5* variant in the Akita population in Japan, investigate its allele frequency, and clarify the contribution of this variant to hair coat variations in Akitas. Furthermore, we aimed to establish a rapid and easy method for genotyping this variant to promote the welfare in this traditional breed through planned breeding.

## 2. Materials and Methods

### 2.1. Animals, Sample Collection, and DNA Extraction

In 2021, saliva, blood, and fur samples were collected from 60 Akitas belonging to the Kagoshima Branch of the AKIHO [20]. This group of 60 dogs was defined as “modern Akitas”. The sample types were determined according to the environment (meeting, show, or kennel) and the dog’s temperament (tame or nervous). The saliva sample was obtained using a swab (Foam Swab, Sterile; Qiagen, Hilden, Germany), impregnated on a filter paper (QIAcard FTA Indicating Classic; Qiagen), and stored in a refrigerator (4 °C) until DNA extraction. The blood sample was obtained via venipuncture, spotted onto a filter paper (QIAcard FTA Classic; Qiagen), and stored in a refrigerator (4 °C) until DNA extraction. The fur sample was obtained by gentle tugging with the fingers. DNA was extracted from discs punched out of the saliva-impregnated and blood-spotted filter papers following appropriate treatments, as previously described [21]. DNA was extracted from the fur using a commercial kit (Isohair; Nippon Gene, Tokyo, Japan), according to the manufacturer’s instructions.

We also used stored blood samples from 73 Akitas collected in the 1970s and the 1980s, which had been stored in a freezer at Azabu University. This group of 73 dogs was characterized as “classic Akitas”. An aliquot of each blood sample was spotted onto a filter paper (QIAcard FTA Classic) and stored in a refrigerator (4 °C) until DNA extraction.

### 2.2. Sanger Sequencing

Sanger sequencing was performed on the coding region in the three exons and exon–intron junctions of the canine *FGF5* gene (Figure 1) using five pairs of primers shown in Table 1. The primers were designed based on the reference sequences of canine *FGF5* mRNA (GenBank accession number NM_001048129.1) and CFA 32 (NC_051836.1, Region 4532937–4554671). Sanger sequencing was carried out on the entire coding region for samples from a long-haired Akita. Additionally, a limited specific region of exon 3, including the c.578C>T position, were sequenced using the Sanger method for samples from two short- and one long-haired Akitas.

Polymerase chain reaction (PCR) was conducted in a 20 µL reaction mixture containing 10 µL 2× PCR master mix (GoTaq Hot Start Green Master Mix, Promega Corp., Madison, WI, USA). The PCR products were purified before sequencing using a QIAquick Gel Extraction Kit (Qiagen), according to the manufacturer’s instructions. Sanger sequencing was performed by Kazusa Genome Technologies, Ltd. (Kisarazu, Japan).

### 2.3. Genotyping by Real-Time PCR for the c.578C>T Variant

The primers and TaqMan minor groove binder probes used for the real-time PCR assay (sequences are listed in Table 1) were designed based on the sequences NM_001048129.1 and CFA 32 (NC_051836.1). These primers and probes, each of which was linked to a fluorescent reporter dye (6-carboxyrhodamine or 6-carboxyfluorescein) at the 5′-end and a non-fluorescent quencher dye at the 3′-end, were synthesized by a commercial company (Applied Biosystems, Foster City, CA, USA). Real-time PCR amplifications were carried out in a final volume of 5 µL consisting of 2× PCR master mix (TaqMan GTXpress Master Mix; Applied Biosystems) and 80× genotyping assay mix (TaqMan SNP Genotyping Assays; Applied Biosystems) containing the specific primers at 450 nM, TaqMan probes at 100 nM, and template DNA. A negative control containing nuclease-free water instead of the template DNA was included in each run. The cycling conditions were 20 s at 95 °C, followed by 50 cycles of 3 s at 95 °C and 20 s at 60 °C, with a subsequent holding stage at 25 °C for 30 s. The data obtained were analyzed using StepOne version 2.3 (Applied Biosystems). Several DNA samples with three different genotypes, wild-type homozygote (C/C), heterozygote (C/T), and mutant homozygote (T/T), from the population of Akitas were used to validate the genotyping assay, following genotype confirmation based on Sanger sequencing.

### 2.4. Scoring Method for Hair Length

Based on apparent hair length, a score was given to 54 modern Akitas (26 males, 26 females, and 2 of unknown sex). A middle-aged female breeder, with a long career specializing in Akita breeding, gave each dog a score based on a scale of 1 (shortest) to 10 (longest) based on her experience alone, without knowledge of the genotyping results. She is not a co-author of this paper and was, therefore, free from any bias in this study.

### 2.5. Statistical Analyses

Statistical analyses were performed using R version 4.5.0. The difference in allele frequencies between modern and classic Akitas was analyzed using Fisher’s exact test. The allele frequencies observed in modern and classic Akitas were analyzed using the chi-square test for Hardy–Weinberg equilibrium (HWE). The deviations between the measured and expected values were analyzed for each group of modern and classic Akitas. Differences in hair length scores among the three genotypes (C/C, C/T, and T/T) were analyzed using the Kruskal–Wallis rank sum test and Wilcoxon rank sum exact test with Bonferroni correction. Differences in hair length scores between males and females were analyzed using the Mann–Whitney U test. Differences at *p* < 0.05 were considered statistically significant.

## 3. Results

### 3.1. Identification of a Variant Causing Long Hair in Akitas

Sanger sequencing of a DNA sample from a long-haired Akita dog revealed that the dog was homozygous (T/T) for the *FGF5*:c.578C>T variant (Figure 3). No other variants or mutations were observed in the coding region of all three exons and the exon–intron junctions. Among short-haired Akitas, some were heterozygous (C/T), and others were wild-type (C/C) (Figure 3).

### 3.2. Genotyping by Real-Time PCR

The newly developed real-time PCR assay (Table 1) clearly differentiated the three genotypes, as determined by Sanger sequencing (Figure 3). Using this real-time PCR assay, we found that the surveyed population of 60 modern Akitas included 21 C/C, 36 C/T, and 3 T/T dogs (Table 2). Based on these observations (C/C, 35.0%; C/T, 60.0%; and T/T, 5%), we estimated a mutant T allele frequency of 0.350. This frequency indicated that the expected percentages of C/C, C/T, and T/T genotypes were 42.3%, 45.5%, and 12.3%, respectively, and chi-square test analysis between the measured and expected data (χ^2^ = 6.0935, df = 2, *p* value = 0.0475) indicated that these three genotypes were not consistent with HWE.

In contrast, among the surveyed population of the 73 classic Akitas, the observed genotype distribution of the dogs was 47 C/C, 21 C/T, and 5 T/T (Table 2). Based on these observations (C/C, 64.4%; C/T, 28.8%; T/T, 6.8%), we estimated a mutant T allele frequency of 0.212. This frequency indicated that expected percentages of C/C, C/T, and T/T genotypes were 62.0%, 33.5%, and 4.5%, respectively, and chi-square test analysis between the measured and expected data (χ^2^ = 1.4302, df = 2, *p* value = 0.4891) indicated that these three genotypes of the observed data were consistent with HWE, unlike the genotypes of modern Akitas.

Furthermore, the observed data between modern and classic Akitas were compared using Fisher’s exact test. The frequency (0.350) of modern Akitas was significantly higher (*p* < 0.001) than that of classic Akitas (0.212) (Table 2).

### 3.3. Scoring of Hair Length

The 10-degree scoring of hair length using 54 modern Akitas revealed that the average scores were 2.78 (range, 1–5) in 18 dogs with the C/C genotype, 4.30 (range, 1–9) in 33 dogs with the C/T genotype, and 8.67 (range, 8–9) in 3 dogs with the T/T genotype (Figure 4). The Kruskal–Wallis rank sum test revealed significant differences among the three genotypes (*p* = 0.0000552). Similarly, the Wilcoxon rank sum exact test with Bonferroni correction revealed significant differences between each pair of genotypes (*p* < 0.05, each *p* value is shown in Figure 4).

In addition, we compared the differences in scores between 26 males and 26 females. The average scores were 4.11 in males and 3.61 in females. There was no significant difference (*p* = 0.223) between the male and female groups, according to the Mann–Whitney U test.

## 4. Discussion

Sanger sequencing of DNA samples from several short- and long-haired Akitas (Figure 2) revealed that a causative variant that determines the long-hair trait in Akitas in Japan was the *FGF5*:c.578C>T variant (Figure 3), as previously reported by researchers from Germany [5]. Long-haired Akitas did not exhibit any other variants or mutations in the coding regions and exon–intron junctions of the *FGF5* gene (Figure 1). All long-haired Akitas examined were homozygous (T/T) for this variant, and all short-haired Akitas examined were either wild-type (C/C) or heterozygous (C/T). Based on these observations, we initially hypothesized that this variant was inherited in an autosomal recessive manner in the Akita population in Japan.

A genotyping survey of 60 modern Akitas using a newly developed real-time PCR assay revealed a mutant T allele frequency of 0.350 (Table 2). This frequency was significantly higher than that obtained from the genotyping survey of 73 classic Akitas (0.212). The three genotypes were not in HWE in modern Akitas but were in HWE in classic Akitas. This deviation was due to a higher-than-expected number of dogs with C/T and a lower-than-expected number of dogs with T/T. This suggests that Akitas with the C/T genotype are more likely to be selected by breeding, and Akitas with T/T genotype (long-hair phenotype) are more likely to be removed from breeding colonies. Based on this, we hypothesized that breeders were more likely to select heterozygous dogs carrying the variant (C/T) based on the appearances of the dogs to breed Akitas with an ideal fluffy hair coat.

Therefore, we performed an experiment to score hair length (Figure 4). There were significant differences among the three genotypes and between each pair of genotypes, indicating that dogs with the C/T genotype were fluffier than those with the C/C genotype. We also examined the differences in hair length scores between male and female Akitas, because dogs with a sex steroid imbalance can have dermatological problems and exhibit truncal hair loss [22]; however, we did not observe a significant difference between them. Thus, these observations suggest that Akita dog breeders were more likely to select heterozygous C/T dogs based on their appearance to breed dogs with an ideal fluffy hair coat.

There was an overlap in hair length scores between C/C and C/T dogs (Figure 4), indicating that the appearance of the hair coat alone was not able to distinguish between these two genotypes. However, many dogs were clearly categorized into these two genotypes based on their apparent hair coat by veteran breeders and ordinary people (Figure 5). These results suggest that the *FGF5*:c.578C>T variant was inherited in an incompletely dominant manner and that the heterozygous C/T genotype exhibited a phenotype with a slightly fluffier hair coat than the wild-type C/C genotype. In recent years, Akita dog breeders have attempted to select heterozygous C/T dogs based on their appearance to breed dogs with an ideal fluffy hair coat according to the changing trends of the Akita dog hair coat.

The *FGF5* gene belongs to a functionally diverse family of growth factors involved in various biological processes, including cell growth, tissue regeneration, embryonic development, metabolism, and angiogenesis [23]. Among *FGF* family members, *FGF1* and *FGF2* genes are known to promote hair growth, whereas *FGF5* acts as an inhibitor of hair elongation [23]. Mice homozygous for the null allele of the *Fgf5* gene, produced by gene targeting in embryonic stem cells, have abnormally long hair [24]. The long hair phenotype appears to be identical to that of mice homozygous for a spontaneous mutation called angora (*go*) [24]. In humans, mutations in the *FGF5* gene cause trichomegaly (MIM 1903309) with long eyelashes, which is inherited in an autosomal recessive manner [25]. Three human mutations, namely c.159_160delTA, c.459+1delG, and c.520T>C, have been reported in three unrelated families with trichomegaly [25,26]. As a defect in the *FGF5* gene can lengthen hair, *FGF5* has attracted attention as a potential therapeutic target for hair loss prevention [27]. Furthermore, *FGF5* has been implicated in cardiovascular function, as it protects the heart from sepsis injury [28]. In humans, variants in the *FGF5* gene are associated with blood pressure regulation and an increased risk of hypertension [29]. To date, no *FGF5*-associated health abnormalities have been reported in animals other than those with long hair; however, in the future, studies may investigate their impacts on cardiovascular function and blood pressure in long-haired individuals.

In the case of Akitas, there is a risk that long-haired dogs may be subjected to treatments that compromise animal welfare, as long hair does not meet the standards of Akitas. Although uncertain, it is reported that long-haired Akita puppies were abandoned or killed in the past. Indeed, in this study, the actual number of long-haired Akitas with the T/T genotype was less than the expected number of T/T dogs, which was calculated based on the high mutant T allele (0.350) in modern Akitas. Some long-haired dogs are suspected of having been removed from breeding colonies, sold or given away to enthusiasts through other distribution channels. The more breeders try to use the variant to create Akitas with an ideal fluffy hair coat, the higher the probability of producing long-haired Akitas, leading to a contradiction. Hence, it is necessary to control the breeding of Akitas by testing the variant, keeping the mutant T allele frequency appropriate, and decreasing the birth rate of long-haired dogs to a level that does not exceed the demand of artists and fanciers, such as in the prevention of canine genetic disorders [17,21,30,31]. This is one strategy used to conserve the traditional Japanese breed and simultaneously ensure animal welfare.

## 5. Conclusions

Our study findings revealed that the *FGF5*:c.578C>T variant determines the long hair trait of Akitas in Japan. This variation is inherited in an incompletely dominant manner. The genotyping survey and hair length scoring by a breeder revealed that Akita dog breeders were more likely to select heterozygous C/T dogs based on the appearances of the hair coat for breeding dogs with an ideal fluffy hair coat. This may result in a high mutant T allele frequency and increased breeding of many long-haired Akitas with the T/T genotype, which may create welfare problems. Planned breeding using genetic tests for this variant is necessary to improve the welfare and conserve this traditional Japanese breed.

## Figures and Tables

**Figure 1 genes-16-00927-f001:**
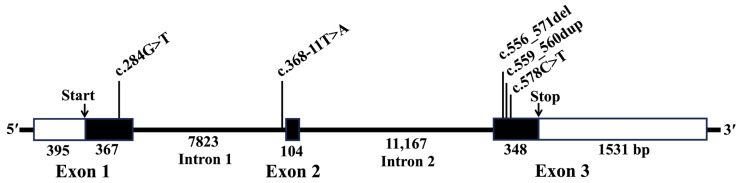
Structure of the canine *FGF5* gene on CFA 32 and locations of five reported variants associated with long hair in dogs. The structure was drawn based on the canine *FGF5* gene transcript 201 (ENSCAFT00845031580.1) in the Ensembl (https://www.ensembl.org/index.html, accessed on 30 July 2025).

**Figure 2 genes-16-00927-f002:**
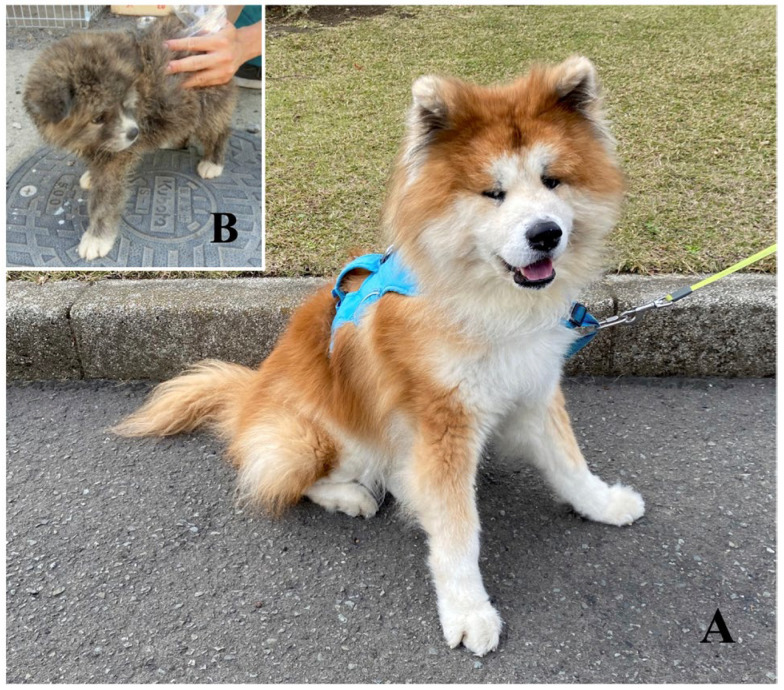
Representative appearances of long-haired Akitas used in this study: (**A**) adult and (**B**) puppy.

**Figure 3 genes-16-00927-f003:**
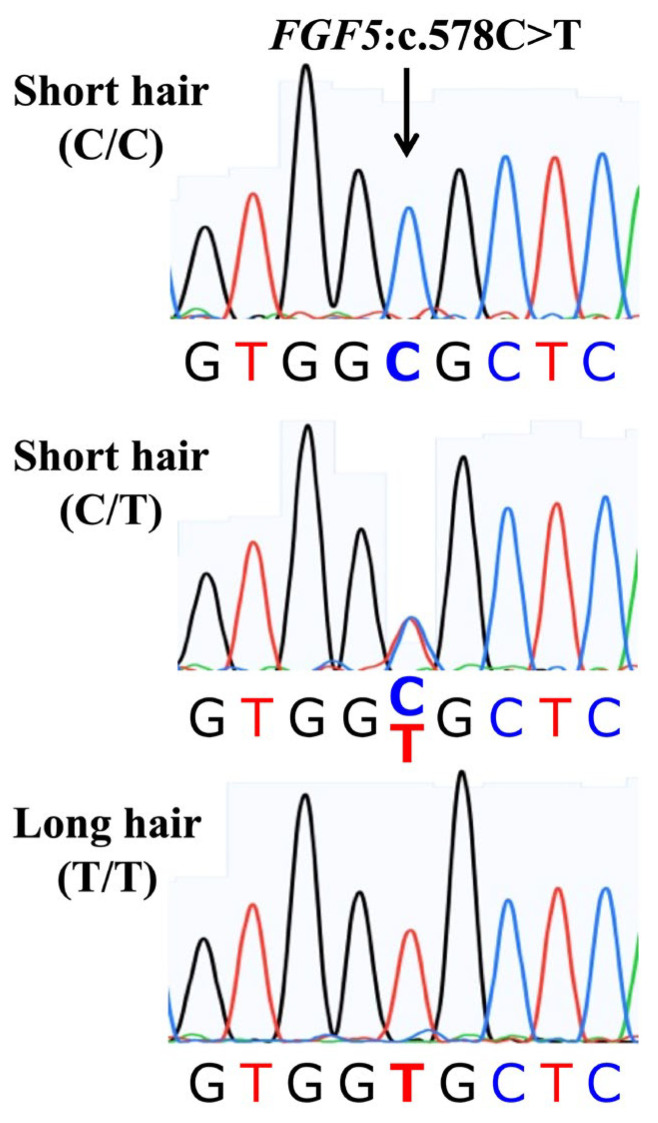
Representative Sanger sequencing electropherograms illustrating the C/C, C/T, and T/T genotypes associated with the canine *FGF5*:c.578C>T variant (arrow). Short-haired Akitas had C/C and C/T genotypes and long-haired Akitas had only T/T genotype.

**Figure 4 genes-16-00927-f004:**
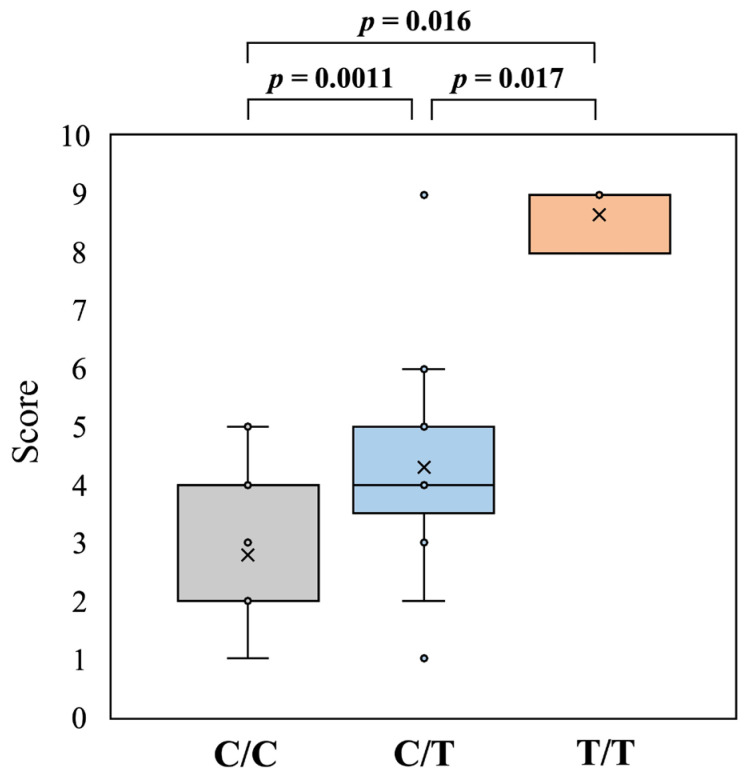
Hair length scores in modern Akitas with three different genotypes (C/C, C/T, and T/T) of the canine *FGF5*:c.578C>T variant. Differences in data between different groups were statistically significant, as revealed by the Kruskal–Wallis rank sum test (*p* = 0.0000552) and Wilcoxon rank sum exact test with Bonferroni correction (each *p* value is shown on the graph).

**Figure 5 genes-16-00927-f005:**
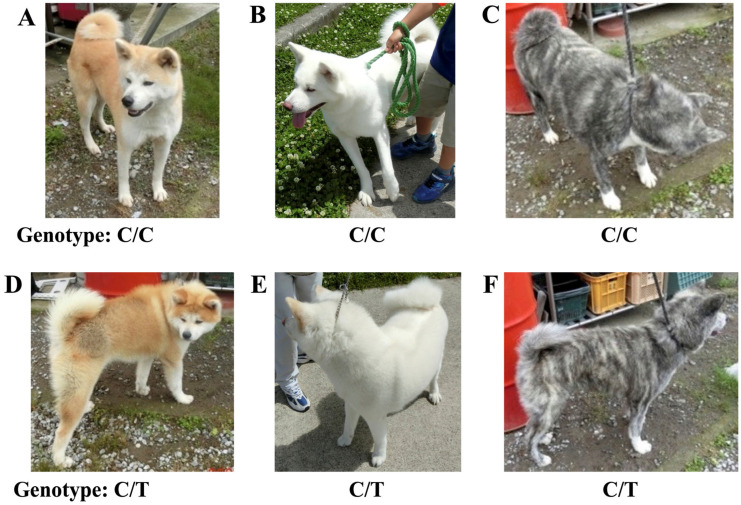
Representative appearance of three major hair-coat colors: red (**A**,**D**), white (**B**,**E**), and brindle (**C**,**F**) in Akitas with C/C genotype (**A**–**C**) and C/T genotype (**D**–**F**) associated with the canine *FGF5*:c.578C>T variant. The hair coat of Akitas with C/T genotype looked slightly fluffier than that of Akitas with C/C genotype.

**Table 1 genes-16-00927-t001:** Sequences of the primers and probes used in the Sanger sequencing and real-time PCR assay for the canine *FGF5* gene and c.578C>T variant.

Primer/Probe	Sequence 5′ to 3′ (mer)	Position (ROS_Cfam_1.0)	Tm (°C) *	Ta (°C) **	Amplicon Size (bp)
Sanger sequencing:					
E1(3)-F	TGGAAGAATGAGCTTGTCCCT (21)	g.4533325_4533345	64.9	58.0	389
E1(3)-R	GCGCGAGCAACTTACTTAAC (20)	g.4533694_4533713	61.5		
E1(4)-F	AGAACCGGCCCTACAAGATG (20)	g.4533286_4533305	65.1	60.0	495
E1(4)-R	AGGGTGCAAAACAACCGCGGTC (22)	g.4533759_4533780	74.6		
E2-F	GCTATAAAGAATGAAAAGAATCTATG (26)	g.4541429_4541454	57.4	60.0	302
E2-R	TCTGAGCCAATTGTTCATCTAAC (23)	g.4541708_4541730	61.9		
E3-F	AGGCCAAGTTTACAGATGACTG (22)	g.4552791_4552812	62.0	61.3	285
E3-R	GAACCTTTGGCTTGACGTGG (20)	g.4553056_4553075	66.7		
E3(2)-F	CCTTTTACCGCAGAAGACCTC (21)	g.4552714_4552734	63.8	60.0	502
E3(2)-R	CTCTTCTGGGAGCTGTAAAG (20)	g.4553196_4553215	58.2		
Real-time polymerase chain reaction:					
Forward primer	CTCCGCAATACACCGAAGTGA (21)	g.4552858_4552878	67.1	60.0	83
Reverse primer	TGCAGCCCCGCTTAGC (16)	g.4552925_4552940	66.5		
Probe for C allele	CTTGTTGAGCGCCACGTA (18)	g.4552915_4552898	63.7		
Probe for T allele	CTTGTTGAGCACCACGTA (18)	g.4552915_4552898	58.2		

* Tm, melting temperature calculated using OlvTools (https://olvtools.com/tmvalue, accessed on 30 July 2025); ** Ta, annealing temperature used in this study. The position of each pair of primers and probes designed in this study were based on Nucleotide BLAST (https://blast.ncbi.nlm.nih.gov/Blast.cgi, accessed on 30 July 2025).

**Table 2 genes-16-00927-t002:** The genotyping survey results of two generational populations of Akitas for the canine *FGF5*:c.578C>T and statistical analyses.

Generational Population	Number of Dogs Examined	Number of Dogs with Each Genotype (%)			Mutant T Allele Frequency	Chi-Square Test
		C/C	C/T	T/T		
Modern Akitas (2021):						
Actual measured data	60	21 (35.0%)	36 (60.0%)	3 (5.0%)	0.350 **	*p* = 0.0475 †
Expected data *		25.4 (42.3%)	27.3 (45.5%)	7.4 (12.3%)		
Classic Akitas (1970s and 1980s):						
Actual measured data	73	47 (64.4%)	21 (28.8%)	5 (6.8%)	0.212 **	*p* = 0.489 ††
Expected data *		45.3 (62.0%)	24.4 (33.5%)	3.3 (4.5%)		

* Expected data were calculated based on the mutant T allele frequency obtained from the actual measured data. ** The difference of mutant T allele frequency between modern and classic Akita populations was analyzed statistically using Fisher’s exact test, resulting in a significant difference (*p* = 0.000833). † Chi-square test analysis (*p* < 0.05) indicated that the three genotypes in the actual measured data of modern Akitas were not in Hardy–Weinberg equilibrium; however, †† Chi-squared test analysis (*p* ≥ 0.05) indicated that the data of classic Akitas were in Hardy–Weinberg equilibrium.

## Data Availability

The original contributions presented in this study are included in the article. Further inquiries can be directed to the corresponding author.

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
