# Peer review of "Genetic Variant of the Canine FGF5 Gene for the Hair Length Trait in the Akita: Utility for Hair Coat Variations and Welfare in Conservation Breeding"

_genes, 2025, doi:10.3390/genes16080927_

Round 1

Reviewer 1 Report

Comments and Suggestions for Authors

The authors present a study genotyping a specific mutation in Akitas and associating it with hair length phenotypes. The mutation and association has been previously established by a prior study.  Normally, confirming the effect of a known mutation in a breed would be unlikely be a sufficiently novel contribution. However, here the authors perform two other analyses that raise the importance of the study. First, they compare allele frequencies in modern and samples from the 1970s/1980s, finding a notable change in the allele frequency.  Second, they present some evidence, although the nature of the data requires some clarification, to argue that C/T heterozygous individuals have a “short and fluffy” phenotype that is currently desired, explaining the persistence of the T allele and the resulting occurrence of T/T long-haired pups from some breedings.  I found these results to be an interesting addition to the FGF5 story, but they would benefit from clearer explanation and a more direct statement about proposed genotype-assisted breeding strategies.

Can the authors add any more information about changes in hair length preferences (particularly the “fluffy” or C/T type?) for Akitas since the 1970s or between Japan and elsewhere?  This would be of interest to the reader and help frame the study.

Methods: The manuscript text should be updated to clearly state how many samples were sequenced?  Was it all 60 modern samples?

Results line 172: to connect  back to the prior work, it might be worthwhile to remind the reader that the 578 variant was previously identified.

Given the modest sample sizes, exact tests for Hardy-Weinberg Equilibrium may be more appropriate than the chi-square test the authors’ apply (see Wigginton, J.E., Cutler, D.J. and Abecasis, G.R. (2005) A note on exact tests of Hardy-Weinberg equilibrium, American Journal of Human Genetics (76)). The authors may wish to consider this approach.  A quick check using available R packages suggests that the conclusions are not altered.

Figure 4 – there appears to be on C/T sample with very long hair.  Have the authors confirmed that this is true and not a sample mix up or genotyping error?

The results of Figure 5 are very interesting and suggest a novel feature to the selection at this locus.  Can the authors provide any more details on the experiment?  How often are the experts “right” in identifying a C/T carrier as the preferred or “fluffy” style? As written, it is somewhat vague.

Discussion/Conclusions: The authors discussion benefits of marker assistant breeding.  Do the authors advocate the breeding pairs be limited to CC x CC or CC x CT?  This would eliminate the TT offspring which seem to be undesirable.  If so, I think the conclusion would be stronger if it was explicitly stated. I believe the proposed idea is that breeding of two “short and fluffy” (i.e., C/T) animals with each other should be avoided to limit the production of the undesired long-haired animals. 

Author Response

Responses to the comments from Reviewer 1

Dear Reviewer 1,

First of all, we appreciate the time the editor and reviewers have taken to read and review our manuscript. Their valuable comments have significantly improved several aspects of our paper. The following document presents our responses to comments and suggestions from the reviewer. It includes the original comments in italics and blue and the subsequent responses we made. The revised parts are indicated in red.

Comments and Suggestions for Authors

The authors present a study genotyping a specific mutation in Akitas and associating it with hair length phenotypes. The mutation and association has been previously established by a prior study.  Normally, confirming the effect of a known mutation in a breed would be unlikely be a sufficiently novel contribution. However, here the authors perform two other analyses that raise the importance of the study. First, they compare allele frequencies in modern and samples from the 1970s/1980s, finding a notable change in the allele frequency.  Second, they present some evidence, although the nature of the data requires some clarification, to argue that C/T heterozygous individuals have a “short and fluffy” phenotype that is currently desired, explaining the persistence of the T allele and the resulting occurrence of T/T long-haired pups from some breedings.  I found these results to be an interesting addition to the FGF5 story, but they would benefit from clearer explanation and a more direct statement about proposed genotype-assisted breeding strategies.

Authors’ response: We appreciate the reviewer's assessment on our study. We are so happy that the reviewer completely understood all of our aims and strengths of our study. Regardless of that, we carefully revised our paper according to the reviewer's comments and suggestions to have our paper improved.

Can the authors add any more information about changes in hair length preferences (particularly the “fluffy” or C/T type?) for Akitas since the 1970s or between Japan and elsewhere?  This would be of interest to the reader and help frame the study.

Authors’ response: Thank you for the reviewer's advice. I think that it is a good point. Indeed, the appearance of Akitas was changed a lot between current and previous ones (the 1970s) when I was a young boy and saw more Akitas at my neighborhood. Akitas' hair coat looked shorter than the current one in my memory. I already checked the standard of Akitas (written in Japanese), but it has not been reissued for a long time. As I mentioned in the Discussion section, "In recent years, Akita dog breeders have attempted to select heterozygous C/T dogs based on their appearance to breed dogs with an ideal fluffy hair coat according to the changing trends of the Akita dog hair coat". I suppose that the trends have gradually been changed from neat short to fluffy short one. None knows the reason. We are sorry that we cannot add any more information about that in our paper.

Methods: The manuscript text should be updated to clearly state how many samples were sequenced?  Was it all 60 modern samples?

Authors’ response: Thank you for your comment and advice. According to the reviewer's advice, we revised a sentence in "2.2. Sanger Sequencing" as follows.

"Sanger sequencing was carried out on the entire coding region for samples from a long-haired Akita. Additionally, a limited specific region of exon 3, including the c.578C>T position, were sequenced using the Sanger method for samples from two short- and one long-haired Akitas."

Results line 172: to connect  back to the prior work, it might be worthwhile to remind the reader that the 578 variant was previously identified.

Authors’ response: Thank you for the reviewer's comment and advice. We added a phase in the Discussion section not in the Result section that the reviewer suggested as follows.

"Sanger sequencing of DNA samples from several short- and long-haired Akitas (Figure 2) revealed that a causative variant that determines the long-hair trait in Akitas in Japan was FGF5:c.578C>T variant (Figure 3), as previously reported by researchers from Germany [5]."

Given the modest sample sizes, exact tests for Hardy-Weinberg Equilibrium may be more appropriate than the chi-square test the authors’ apply (see Wigginton, J.E., Cutler, D.J. and Abecasis, G.R. (2005) A note on exact tests of Hardy-Weinberg equilibrium, American Journal of Human Genetics (76)). The authors may wish to consider this approach.  A quick check using available R packages suggests that the conclusions are not altered.

Authors’ response: Thank you for the reviewer’s valuable advice. We also checked HWE exact test and obtained p = 0.0022 for modern Akitas and p = 0.287 for classic Akitas, suggesting almost same results as chi-square test. We are relieved to know these results.

We have used chi-square test for this type of analysis for a long time. We would like to use a exact test for HWE next time. This time, if we used both methods in our paper, the data would be too complicated for readers to understand. Therefore, we would like to omit a use of the exact test because the conclusion is not altered. Thank you for your understanding.

Figure 4 – there appears to be on C/T sample with very long hair.  Have the authors confirmed that this is true and not a sample mix up or genotyping error?

Authors’ response: Thank you for the reviewer’s comment. That is a good point that you pointed out. This is a genotyping error by a breeder (she mistook this dog for a long-hair Akita) based on the appearance because this dog was a puppy probably at 2 or 3 weeks of age. Some puppy sometimes looks very fluffy like long-haired dogs when they are under 3 weeks of age. The long-hair phenotype becomes obvious at 4 weeks of age, which a breeder does not mistake to identify. We used this data (human error) because a breeder had to be free from any bias. We did not ask why she mistook this dog and how old that dog was exactly. So, we did not explain this outlier. We appreciate your understanding.

The results of Figure 5 are very interesting and suggest a novel feature to the selection at this locus.  Can the authors provide any more details on the experiment?  How often are the experts “right” in identifying a C/T carrier as the preferred or “fluffy” style? As written, it is somewhat vague.

Authors’ response: Thank you for the reviewer's important comment. We just performed this kind of scoring once so far. As we mentioned in the Discussion section, "There was an overlap in hair length scores between C/C and C/T dogs (Figure 4), indicating that the appearance of the hair coat alone was not able to distinguish between these two genotypes." After this paper's publication, we are going to contact the AKIHO and we would to have scoring testing together with many breeders. Furthermore, we are going to establish a rule for breeding based on the genotyping results. Those activities and experiments will clarify the unknown aspects the reviewer pointed out.

Discussion/Conclusions: The authors discussion benefits of marker assistant breeding.  Do the authors advocate the breeding pairs be limited to CC x CC or CC x CT?  This would eliminate the TT offspring which seem to be undesirable. If so, I think the conclusion would be stronger if it was explicitly stated. I believe the proposed idea is that breeding of two “short and fluffy” (i.e., C/T) animals with each other should be avoided to limit the production of the undesired long-haired animals.

Authors’ response: Thank you for your valuable comment. We have not advocated a clear criteria or rule for breeding based on the genotypes so far. We will contact the AKIHO, investigate the current status, hear the opinions of many breeders, and establish the rule and criteria. That is an ideal process. We decided not to write a clear and simple method in our paper so far because an ideal rule and criteria will be established after collaboration and negotiation with the AKIHO. We appreciate your understanding.

Reviewer 2 Report

Comments and Suggestions for Authors

Genetic Variant of the Canine FGF5 Gene for the Hair Length Trait in the Akita: Utility for Hair Coat Variations and Welfare in Conservation Breeding

Dear Authors,

The manuscript is very interesting and very well prepared. Statistical analysis and obtained results confirmed that the genetic variant FGF5:c.578C>T gene is not inherited in a non-dominant manner, and genetic testing and breeding planning will certainly help limit the appearance of long-haired Akitas from short-haired parents (alternatively, one could also try to plan a system for the adoption of long-haired Akitas puppies by private owners after rearing with reimbursement of expenses during this period).

Below I added several suggestions helpful in revision process:

Line 85

Maybe better is to move Figure 1 with title in Introduction section after first paragraph to the line 50, because in line 54 mention about it is added.

Line 130

In Table 1 in header please add one asterisk to Tm (°C)* and two in case of Ta (°C)**.

Line 131

Please add one and two asterisks respectively to Tm and Ta according to header signature.

Lines 188 and 204

p-value

Lines 262-265

One-two references can be added in case of this sentence.

Lines 262-265

One reference can be added here.

Author Response

Responses to the comments from Reviewer 2

Dear Reviewer 2,

First of all, we appreciate the time the editor and reviewers have taken to read and review our manuscript. Their valuable comments have significantly improved several aspects of our paper. The following document presents our responses to comments and suggestions from the reviewers. It includes the original comments in italics and blue and the subsequent responses we made. The revised parts are indicated in red.

Comments and Suggestions for Authors

Genetic Variant of the Canine FGF5 Gene for the Hair Length Trait in the Akita: Utility for Hair Coat Variations and Welfare in Conservation Breeding

Dear Authors,

The manuscript is very interesting and very well prepared. Statistical analysis and obtained results confirmed that the genetic variant FGF5:c.578C>T gene is not inherited in a non-dominant manner, and genetic testing and breeding planning will certainly help limit the appearance of long-haired Akitas from short-haired parents (alternatively, one could also try to plan a system for the adoption of long-haired Akitas puppies by private owners after rearing with reimbursement of expenses during this period).

Authors’ response: We appreciate the reviewer's assessment on our study. We are so happy that the reviewer completely understood all of our aims and strengths of our study. We also thank for your offering a future system. I agree with your opinion.

After our paper is published, we will contact the AKIHO, investigate the current status in a larger scale, hear the opinions of many breeders, and establish the rule and criteria to decrease the birth of long-haired Akitas and simultaneously keep a fluffy hair coat with C/T genotype. That is an ideal process. We decided not to write a clear and simple method in our paper so far because an ideal rule and criteria will be established after collaboration and negotiation with the AKIHO.

Below I added several suggestions helpful in revision process:

Line 85

Maybe better is to move Figure 1 with title in Introduction section after first paragraph to the line 50, because in line 54 mention about it is added.

Authors’ response: Thank you for the reviewer's advice. We agree with your opinion, but we moved Figure 1 after the second paragraph (line 62) because Figure 1 was first mentioned in the second paragraph. Thank you.

Line 130

In Table 1 in header please add one asterisk to Tm (°C)* and two in case of Ta (°C)**.

Line 131

Please add one and two asterisks respectively to Tm and Ta according to header signature.

Authors’ response: Thank you for the reviewer's advice. We revied these according to the reviewer's suggestion.

Lines 188 and 204

p-value

Authors’ response: Thank you for finding our mistakes. We corrected these.

Lines 262-265

One-two references can be added in case of this sentence.

Lines 262-265

One reference can be added here.

Authors’ response: Thank you for the reviewer's valuable advice. We received a similar advice from another reviewer regarding this issue. Indeed, the appearance of Akitas was changed a lot between current and previous ones (the 1970s) when I was a young boy and saw more Akitas at my neighborhood. Akitas' hair coat looked shorter than the current one in my memory. I already checked the standard of Akitas (written in Japanese), but it has not been reissued for a long time. As I mentioned in the Discussion section (a part the reviewer pointed out), "In recent years, Akita dog breeders have attempted to select heterozygous C/T dogs based on their appearance to breed dogs with an ideal fluffy hair coat according to the changing trends of the Akita dog hair coat". I suppose that the trends have gradually been changed from neat short to fluffy short one. None knows the reason. We are sorry that we cannot add any more information (references) about that in our paper.

Furthermore, we will contact the AKIHO, investigate the current status, hear the opinions of many breeders, and establish the rule and criteria. That is an ideal process. We decided not to write a clear and simple method in our paper so far because an ideal rule and criteria will be established after collaboration and negotiation with the AKIHO. We appreciate your understanding.